# GarmentPainter: Efficient 3D Garment Texture Synthesis with Character-Guided Diffusion Model

## Abstract

Generating high-fidelity, 3D-consistent garment textures remains a challenging problem due to the inherent complexities of garment structures and the stringent requirement for detailed, globally consistent texture synthesis. Existing approaches either rely on 2D-based diffusion models, which inherently struggle with 3D consistency, require expensive multi-step optimization or depend on strict spatial alignment between 2D reference images and 3D meshes, which limits their flexibility and scalability. In this work, we introduce *GarmentPainter*, a simple yet efficient framework for synthesizing high-quality, 3D-aware garment textures in UV space. Our method leverages a UV position map as the 3D structural guidance, ensuring texture consistency across the garment surface during texture generation. To enhance control and adaptability, we introduce a type selection module, enabling fine-grained texture generation for specific garment components based on a character reference image, without requiring alignment between the reference image and the 3D mesh. GarmentPainter efficiently integrates all guidance signals into the input of a diffusion model in a spatially aligned manner, without modifying the underlying UNet architecture. Extensive experiments demonstrate that GarmentPainter achieves state-of-the-art performance in terms of visual fidelity, 3D consistency, and computational efficiency, outperforming existing methods in both qualitative and quantitative evaluations.

## 1 Introduction

The demand for realistic 3D garment visualization is increasing across e-commerce, gaming, 3D films, and AR/VR, driving the need for high-quality garment assets. While 2D image generation Chen et al.; Rombach et al. (2022); Peebles & Xie (2023); Sauer et al. (2022); Wang et al. (2021) has been greatly advanced by generative AI, producing high-fidelity 3D garment textures remains labor-intensive, often requiring skilled artists and weeks of work. This underscores the need for efficient, automated 3D garment generation methods.

Since garments are worn directly on the human body, texture quality is crucial for realism and aesthetic appeal. High-fidelity textures not only enhance visual authenticity but also elevate the wearer's presence and delivering a refined aesthetic. Although existing methods Zeng et al. (2024); Wu et al. (2024); Yu et al. (2024; 2023); Kingma (2013); Zhang et al. (2024b) have made remarkable progress, generating high-quality garment textures remains challenging. This difficulty arises from three primary factors: **Garment-Aware Generation.** Reference images provide rich, fine-grained cues for texture generation and have shown strong effectiveness in prior work Zeng et al. (2024); Yu et al. (2023); Liu et al. (2024). Leveraging natural person images as references is especially appealing for garment texture generation, as it captures appearance in realistic contexts and enables synthesizing multiple garment components from a single image. However, existing approaches Zeng et al. (2024); Yu et al. (2023); Bensadoun et al. (2024); Yu et al. (2024) rely on precise alignment between the reference and the garment mesh, and misalignment often leads to irrelevant textures and degraded visual quality. (As shown in Figure 6.) **3D Consistency.** Most existing texturing methods Zeng et al. (2024); Yu et al. (2023); Chen et al. (2023a); Wu et al. (2024) use Latent Diffusion Models (LDMs) Rombach et al. (2022) to generate multi-view images, which are then projected onto 3D assets to create textures. While pre-trained 2D diffusion models allow for diverse texture gener-

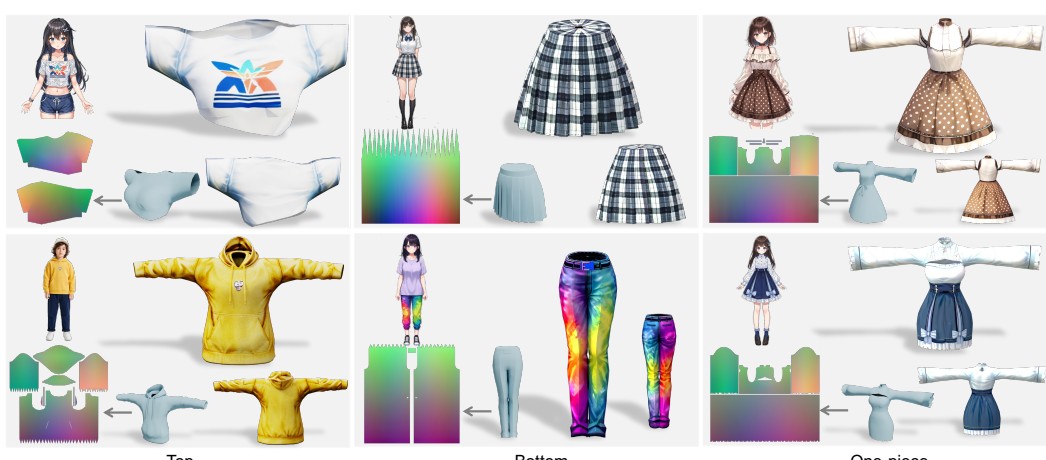

| Top | Bottom | One-piece |

Figure 1: The results of our method. Given garment type, we showcase three types of garment texture generation: *top, bottom and one-piece*. Each image consists of two sections: The left side displays the reference image alongside the UV position map, which is obtained through the rasterization of the corresponding mesh. The right side presents the rendered output of the generated texture, including both the front view and back view. Our method produces high-quality and 3D consistent textures across a wide range of garment styles.

ation, they struggle with 3D consistency due to inherent model limitations. This often results in viewpoint-dependent inconsistencies, causing visual artifacts and unnatural transitions on the mesh surface. In certain perspectives, the model fails to maintain accurate textures, leading to noticeable 3D distortions. **Generation Efficiency.** Previous methods typically use multi-step processes Zhang et al. (2024b); Richardson et al. (2023); Yu et al. (2023); Zeng et al. (2024); Bensadoun et al. (2024) or per-instance optimization Metzer et al. (2023) to achieve high-quality results. However, these approaches are both computationally expensive and time-consuming. Moreover, integrating diverse information at different stages requires specialized modules, adding further overhead and slowing down the process.

The limitations discussed above hinder the performance of existing garment texture generation methods. In this work, we introduce **GarmentPainter**, a simple and efficient framework for generating high-quality garment textures. To support this, we organize a dedicated garment texture dataset, where each sample includes a reference image, UV texture map, garment type label, and UV position map. The UV position map preserves the layout structure of the UV texture map while retaining surface position information of the 3D garment. More details are provided in Sec. 3. Furthermore, we design a diffusion model that effectively synthesizes garment textures in UV space. A character image serves as style guidance, while the UV position map ensures 3D consistency. We extract latent representations from both the reference image and UV position map as guidance signals. These latents are spatially aligned with the diffusion input noise, enabling efficient information sharing and ensuring that the generated textures are accurate and contextually relevant. To enhance control, we introduce a type selection module that encodes garment type labels and integrates them with the time embedding via a residual connection. This allows for precise texture generation based on specified garment components (*i.e.*, top, bottom, or one-piece) while preventing the introduction of irrelevant textures and eliminating the need for strict alignment between the reference image and garment mesh. By employing an efficient information injection strategy, we enable effective information interaction without modifying the diffusion transformer. This design preserves generation efficiency, ensuring that texture synthesis remains as fast as standard image generation.

Our key contributions are as follows: 1) We construct a high-quality garment dataset specifically for garment texture generation. 2) We introduce GarmentPainter, a simple yet effective framework that generates high-fidelity, 3D-consistent textures, accurately aligning with the specified garment regions in the reference image. 3) Our method leverages unified UV unfolding to align 2D images

with 3D texture generation, enabling 2D models to synthesize 3D UV textures more effectively and providing new insights. 4) Extensive qualitative and quantitative evaluations demonstrate that our method outperforms state-of-the-art approaches in garment texture generation.

## 2 RELATED WORKS

**2D Virtual Try-ON** 2D virtual try-on has made significant progress based on 2D diffusion models. This task aims to transfer garments from one image to another, given a clothing template or character image. To handle different types of garments, existing approaches tend to train specialized models that individually process various clothing categories. Several works leverage the capabilities of diffusion models to extract features from reference images. For instance, OOTDiffusion Xu et al. (2024) and IDM-VTON Choi et al. (2024) construct new branches within diffusion architectures to process reference images and inject conditional information at different timesteps with corresponding latent representations. CatVTON Chong et al. employs a single-stream approach to simultaneously perform generation and feature processing tasks, proving to be an efficient methodology for virtual try-on. However, the results of these methods cannot be readily applied in spatial contexts. Extending virtual try-on to 3D environments and unifying the generation process across different garment types is crucial for 3D applications and spatial tasks.

**3D Generation via 2D generation model.** With the rise of text-to-image generation techniques, numerous 3D texturing methods have begun leveraging 2D generative models to improve 3D generation optimization. Prior optimization-based approaches Hong et al. (2022); Chen et al. (2022a); Ma et al. (2023); Michel et al. (2022); Mohammad Khalid et al. (2022) primarily focused on refining the texture maps of 3D models through large-scale vision-language models (e.g., CLIP Radford et al. (2021)), predating the advent of LDMs. DreamFusion Poole et al. introduced Score Distillation Sampling (SDS) to 3D generation, laying the groundwork for many subsequent text-to-3D methods. Building on this, Latent-nerf Metzer et al. (2023) and Fantasia3D Chen et al. (2023b) extended SDS for texture mapping. Recently, TexPainter Zhang et al. (2024b) performs multi-view fusion in the 2D color space to address inconsistency issues. However, optimization-based methods often require lengthy training periods. As an alternative, TEXTure Richardson et al. (2023), Text2Tex Chen et al. (2023a), Garment3DGen Sarafianos et al. and Paint3D Zeng et al. (2024) directly generate texture images from multiple 3D mesh views and then paint them onto the meshes. TEXTure Richardson et al. (2023) and Text2Tex Chen et al. (2023a) adopt this concept, continually modifying overlapping segments of the texture image from the previous view with each new camera pose. However, these approaches often suffer from artifacts such as seams, noise, and irrelevant fragments. To address these challenges, Paint3D Zeng et al. (2024) introduces a specialized model designed to fill incomplete regions during the texture painting process. FabricDiffusion Zhang et al. (2024a) generates local patterns with a 2D diffusion model and synthesizes textures by tiling and overlay, which struggles with complex textures.

**Generative Texturing from 3D Data.** Furthermore, generative texture models can be trained from scratch using 3D data. Early approaches Oechsle et al. (2019); Gao et al. (2022); Xiong et al. (2024) relied on implicit texture fields to assign color to each surface pixel of a 3D shape. However, because the texture on a 3D mesh is continuous, purely discrete supervision often falls short of producing high-fidelity textures.

Texturify Siddiqui et al. (2022) tackles this challenge by constructing a texture map on the surface of a polygonal mesh, while Mesh2tex Bokhovkin et al. (2023) builds on this by integrating an implicit texture field for enhanced results. TexOct Liu et al. (2024) employs point clouds to handle the variability introduced by diverse mesh topologies and UV mappings. Other methods Chen et al. (2022b); Yu et al. (2023) synthesize UV maps for 3D meshes but struggle to accommodate generic objects due to the inherent diversity found among different 3D mesh categories.

## 3 GARMENT DATASET

**Dataset Description.** We construct a comprehensive dataset from VRoidHub, following the guidelines of Panic3D Chen et al. (2023c), encompassing a diverse range of character models. Each model in VRoidHub consists of body, garment, and animation components. Using Blender, we segment garments into distinct parts based on their components. After segmentation, we refine the dataset by

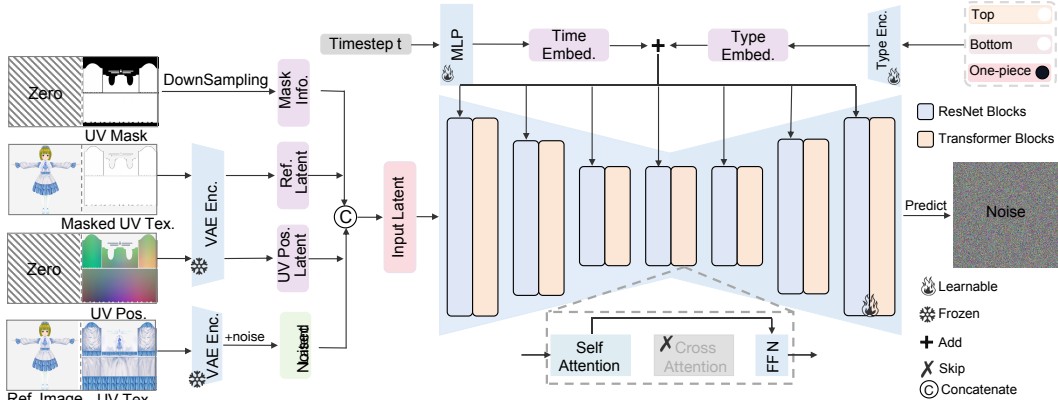

Figure 3: The GarmentPainter framework starts by encoding the reference image and UV position map with a VAE Kingma (2013). At the same time, the masked UV texture map is encoded to preserve background regions that should not be regenerated. The reference latent captures style information, while the UV position latent captures structure. These latents, along with the UV texture map and reference image, are transformed into texture latents, then noised to form the noisy latent. The UV mask image is downsampled by 8× to restrict generation to the intended area. All latent features are concatenated and passed into the diffusion model. A type encoder also embeds garment type labels, which are added to the time embedding to provide fine-grained control over garment regions.Note: Each transformer block omits cross-attention for text interaction.

removing samples with inconsistent color distributions, monochromatic designs, or garments featuring accessories such as ties, belts, or other extraneous elements. For garment classification, we first render each mesh from two views, *i.e.* front and back, and concatenate them into a single image. We then leverage GPT-4oAchiam et al. (2023) to classify garment types using the following prompt:

*"What type does this image belong to? Choose from [Top, Bottom, Onepiece]."*

along with the rendered image. All labels undergo verification by human reviewers. We discard data where garments are completely occluded but retain samples that are only partially occluded by other clothing to enhance model robustness. Ultimately, we curate a dataset comprising 7,579 clothing items, including 3,703 tops, 2,114 bottoms, and 1,762 one-piece garments.

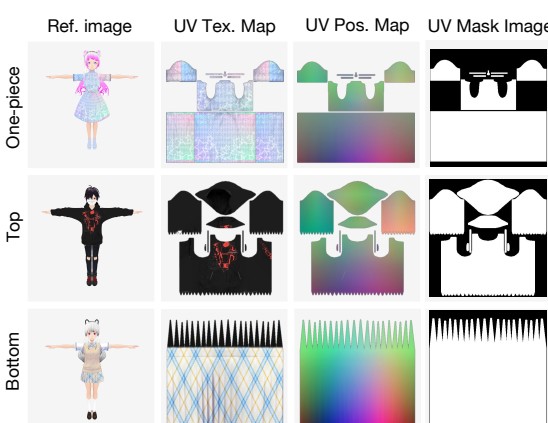

Figure 2: We show three types of our data, *i.e.*, *one-piece, top* and *bottom*. Specifically, we show reference image, UV texture map, UV position map and UV mask image.

**Rasterization.** In a UV texture map, the RGB information of the surface mesh is stored, and during rendering, a rasterization operator samples the color from the UV texture image onto the rendered image. A UV position map stores the XYZ coordinates of the mesh surface, which requires sampling mesh properties and baking them into the UV map. Please refer A.1 for more details. We render all garment data and character data from two perspectives: front and back. All images are generated using Nvdiffrast Laine et al. (2020) under uniform ambient lighting. An illustrative example is shown in Figure 2.

## 4 METHOD

Given a 3D garment mesh with an initial UV mapping and a front-view reference image of a character, our objective is to generate a high-quality UV texture map that ensures 3D consistency while preserving fine-grained details aligned with the reference. To achieve this, we introduce Garment-Painter, a framework built upon a pre-trained inpainting diffusion model Rombach et al. (2022). Since text conditions are not required for our method, we remove the text encoder and cross-attention modules from the UNet to streamline the model architecture. As illustrated in Figure 3, our approach leverages the following components: 1) A reference image, which provides garment style information for the generated garment texture. 2) A UV position map, which encodes structural information, capturing the basic UV texture layout and the 3D coordinates of points on the garment mesh surface. GarmentPainter employs a VAE Kingma (2013) encoder to process these inputs, encoding the reference image and UV position map into reference latent and UV position latent, respectively. These latent representations are then concatenated with the noised latent and mask information before being fed into the diffusion model. To facilitate precise and flexible control over the generation of upper and lower garment parts, we introduce a type selection module. This module encodes type label information to generate type embeddings, which are subsequently combined with time embeddings as an additional control signal. By injecting this information into the diffusion model, our approach enables fine-grained control over the texture generation for the top, bottom, or one-piece garment texture. We further elaborate on this process in the following sections: *Reference Condition Injection, 3D Information Injection*, and *Type Selection*.

### 4.1 REFERENCE CONDITION INJECTION

In our method, we use an image of a character wearing clothing as the reference, ensuring that the generated garment texture aligns with the one in the reference. A character image serves as a reference, providing two advantages. First, it is more flexible and user-friendly, as garments are typically viewed in context with a character rather than in isolation. As a result, such images are more natural, abundant, and easily accessible compared to standalone garment images, facilitating a more intuitive and practical reference selection. Second, it eliminates the need for additional complex modules, such as image segmentation for garment extraction, thereby reducing computational overhead and parameter complexity. This makes our approach both more efficient and streamlined.

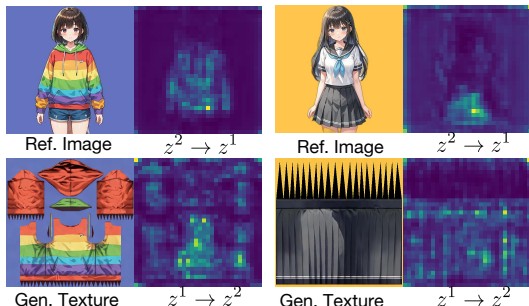

Figure 4: Attention maps for $z^2 \rightarrow z^1$ and $z^1 \rightarrow z^2$ demonstrate accurate and effective information exchange between the two diffusion noise predictors $\epsilon_\theta^1, \epsilon_\theta^2$

To achieve both controllability and parameter efficiency, we utilize a VAE Kingma (2013) encoder $\mathcal{E}$ encoding the reference image $\mathcal{I}_{\text{ref}}$ into latent feature representation $\mathcal{F}_{\text{ref}} = \mathcal{E}(\mathcal{I}_{\text{ref}}) \in \mathbb{R}^{C \times H \times W}$. Previous studies Wang & Shi (2023); Shi et al. (2023) have also demonstrated that this latent feature effectively preserves local detailed information. During training process, we organize the reference part of diffusion input as follows:

$$\mathcal{F}_{\text{x}} = \text{AddNoise}_t([\mathcal{Z}_{\text{uv}} \oplus \mathcal{F}_{\text{ref}}]) \copyright [\mathcal{Z}_{\text{m}} \oplus \mathcal{F}_{\text{ref}}] \tag{1}$$

where $\mathcal{Z}_{\text{uv}} \in \mathbb{R}^{C \times H \times W}$ is the latent of UV texture image $\mathcal{I}_{\text{uv}}$ encoded by the $\mathcal{E}$. $\mathcal{Z}_{\text{m}} = \mathcal{E}(\mathcal{I}_{\text{uv}} \cdot \mathcal{I}_{\text{mask}})$ is the masked image's latent of $\mathcal{I}_{\text{uv}}$ and correspond mask image $\mathcal{I}_{\text{mask}}$. $\copyright$ denotes the concatenation in the the channel dimension. $\oplus$ denotes the concatenation in the spatial dimension. $\text{AddNoise}_t$ means forward process in diffusion with a timestep $t$. The result of $\mathcal{F}_{\text{x}}$ is with shape of $2C \times H \times 2W$. During the inference process, we replace the $[\mathcal{Z}_{\text{uv}} \oplus \mathcal{F}_{\text{ref}}]$ with a pure noise $[\eta_{\text{uv}} \oplus \eta_{\text{ref}}]$. The $\mathcal{F}_{\text{x}}$ is formulated as

$$\mathcal{F}_{\text{x}} = [\eta_{\text{uv}} \oplus \eta_{\text{ref}}] \copyright [\mathcal{Z}_{\text{m}} \oplus \mathcal{F}_{\text{ref}}] \tag{2}$$

**Analysis of the Reference Image Guidance Mechanism.** At each timestep $t$, we employ two noise predictors, $\{\epsilon_\theta^1, \epsilon_\theta^2\}$, corresponding to the reference noise $\eta_{\text{ref}}$ and the UV noise $\eta_{\text{uv}}$. Both predictors estimate the noise of the latent $z_t$ to obtain the next latent state $z_{t-1}$. In our approach, the

prediction of $\epsilon_\theta^2$ depends not only on its own latent state $z_t^2$ but also on the latent state $z_t^1$ produced by $\epsilon_\theta^1$. In practice, the two predictors share the same UNet architecture, and their input latents are concatenated so that information can be exchanged through the self-attention layers without requiring any additional modules. Consequently, $\epsilon_\theta^2$ can be interpreted as a conditional noise predictor: $\epsilon_\theta^2(z_t^2, t) = \epsilon_\theta(z_t^2; t, z_t^1)$, and symmetrically, $\epsilon_\theta^1(z_t^1, t) = \epsilon_\theta(z_t^1; t, z_t^2)$. As illustrated in Figure 4, our method establishes bidirectional correspondences between the reference image and the generated texture. For the " Top" garment (left), the attention map $z^2 \to z^1$ highlights the hoodie region in the reference image, while the reverse attention $z^1 \to z^2$ focuses on the corresponding area in the generated UV texture. A similar effect is observed for the "Bottom" garment (right). These results demonstrate that our method is *Garment-aware*, automatically localizing garment-specific regions, while the two noise predictors efficiently and accurately exchange information under our design.

## 4.2 3D STRUCTURE INFORMATION INJECTION

In addition to ensuring that the generated garment texture aligns with the reference image, maintaining 3D consistency is crucial. In our method, we employ a UV representation to unwrap the 3D surface onto a 2D plane. This process maps each point on the 3D model to a specific location in 2D space, ensuring that every detail of the 3D surface accurately corresponds to a position on the UV map. However, adjacent regions in 3D space may become non-adjacent in the UV map, leading to a loss of global 3D structural information. To address this issue, we introduce a UV position map $\mathcal{I}_{pos}$ as 3D structural guidance. As described in Sec. 3, $\mathcal{I}_{pos}$ shares the same layout as the UV texture map $\mathcal{I}_{uv}$ but encodes the 3D position (XYZ) coordinates of the mesh within a unified coordinate system. This effectively embeds 3D spatial information into a 2D representation, preserving structural relationships from the original 3D space and ensuring global consistency in the generated textures.

A common approach to incorporating this information is to introduce an additional module to encode $\mathcal{I}_{pos}$ and embed the encoded representation into the diffusion model. However, this adds extra parameters and increases computational complexity. Instead, we inject 3D structural information by encoding $\mathcal{I}_{pos}$ through a VAE Kingma (2013) encoder, obtaining a latent representation $\mathcal{F}_{pos} \in \mathbb{R}^{C \times H \times W}$. This latent feature is then concatenated with $\mathcal{F}_x$ as input to the diffusion model. Our approach is based on an experimental observation: although the UV position map is not a conventional image, it can still be reconstructed with minimal loss after passing through the VAE Kingma (2013) encoder and decoder. This suggests that once encoded into a latent representation, it retains its original structural information (More details are provided in supplementary material). The process is formulated as:

$$\mathcal{F}_y = \mathcal{F}_x \copyright [\mathcal{F}_{pos} \oplus \mathcal{Z}_{zero}], \tag{3}$$

where $\mathcal{Z}_{zero}$ is a zero-padding tensor of the same shape as $\mathcal{F}_{pos}$, ensuring dimensional alignment with $\mathcal{F}_x$. This method offers two advantages. First, it only requires adjusting the input convolution channels of the diffusion model to integrate the new input into the U-Net, ensuring efficient parameter utilization. Second, it improves inference efficiency without significantly increasing computational overhead.

## 4.3 TYPE SELECTION

While the UV position map inherently encodes outfit type information, variations and discrepancies between the outfit in a character's image and the given asset mesh require explicit control for greater flexibility and robustness. To address this while preserving parameter efficiency and inference speed, we introduce a type selection module.

We define a three-class one-hot vector $\mathcal{S}_{cls}$ based on the garment type label. Using a learnable embedding function, we map $\mathcal{S}_{cls}$ to a signal representation that matches the dimensionality of the diffusion time embedding $\mathcal{T}_{emb}$. This signal is then combined with $\mathcal{T}_{emb}$ to inject garment part control information into the diffusion process:

$$\mathcal{F}_{cls} = \text{ProjectEmb}(\text{PosEmb}(\mathcal{S}_{cls})) + \mathcal{T}_{emb}, \tag{4}$$

where ProjectEmb is a two-layer MLP, and PosEmb applies positional encoding, similar to Long et al. (2024). We replace the original $\mathcal{T}_{emb}$ in the diffusion model with $\mathcal{F}_{cls}$, seamlessly integrating the control signal into the diffusion process. This method introduces minimal additional parameters while enabling precise control over the generation of specific garment parts.

### 4.4 TRAINING

Finally, we organize the input of our model during the training process as follows:

$$\mathcal{F}_{\text{in}} = \mathcal{F}_{\text{y}} \copyright [\text{Downsample}(\mathcal{I}_{\text{mask}}) \oplus \mathcal{Z}_{\text{zero}}], \tag{5}$$

where the mask $\mathcal{I}_{\text{mask}}$ is downsampled to match the spatial shape with a single channel, and $\mathcal{Z}_{\text{zero}}$ is a zero-padding tensor of the same shape Downsample($\mathcal{I}_{\text{mask}}$). Our objective function is then formulated as:

$$\mathcal{L}_{LDM} = \mathbb{E}_{\epsilon \sim \mathcal{N}(0,1), t}$$
$$\left[ \| \epsilon - \epsilon_\theta([\mathcal{Z}_{\text{uv}} \oplus \mathcal{F}_{\text{ref}}], t, (\mathcal{I}_{\text{ref}}, \mathcal{I}_{\text{pos}}, \mathcal{I}_{\text{mask}}, \mathcal{S}_{\text{cls}})) \|_2^2 \right] \tag{6}$$

## 5 EXPERIMENTS

**Implementation Details.** We apply the inpainting model from Stable Diffusion v1.5 Rombach et al. (2022) as our texture generation backbone. We trained the all UNet parameters expect the cross attention modules with a constant learning rate $1e-5$, we train our module $70,000$ steps with 7 NVIDIA-A100s GPU with batch size 15 per-GPU.

**Baselines.** We compare our method with several state-of-the-art approaches, including TEX-Ture Richardson et al. (2023), Text2Tex Chen et al. (2023a), Paint3D Zeng et al. (2024) and TEX-Gen Yu et al. (2024). TEXTure Richardson et al. (2023) introduces an iterative texture generation strategy to enhance texture manipulation. Text2Tex Chen et al. (2023a) builds on this by incorporating an automatic viewpoint selection process within each iteration. Paint3D Zeng et al. (2024) is a coarse-to-fine texture synthesis approach. First, it samples multi-view images from pre-trained 2D diffusion models, then refines the textures in UV space using a diffusion model conditioned on both the position map and the coarse texture. TEXGen Yu et al. (2024) synthesizes detailed and coherent textures, leveraging a novel hybrid 2D-3D block that adeptly manages both local detail fidelity and global 3D-aware interactions.

**Evaluation metrics**. We utilize commonly used metrics for evaluating the performance of generative models, specifically, the Fréchet Inception Distance (FID) Heusel et al. (2017) and Kernel Inception Distance (KID) Bińkowski et al. (2018), which measure image quality and image diversity. Additionally, we also evaluate the texture generation speed.

### 5.1 COMPARISON WITH STATE-OF-THE-ART METHODS

In this section, we compare our method with state-of-the-art methods both qualitatively and quantitatively. State-of-the-art methods require different types of inputs. To ensure a fair comparison, we carefully configure the inputs for each approach, as detailed in A.2. Notably, only Paint3D Zeng et al. (2024) can support character image as input with unknown pose.

**Qualitative comparisons.** In Figure 5, we present a qualitative comparison of our method with state-of-the-art approaches. Despite operating in a more complex setting, our method generates garment textures that exhibit enhanced 3D consistency while remaining well-aligned with the reference image. The strongest competitor, TEXGen Yu et al. (2024), renders its first view using ground-truth UV texture mapping, effectively capturing fine details from the reference image. However, its second view often appears rough and lacks 3D consistency. In contrast, even when provided with a character image containing irrelevant background content, our method consistently produces high-quality texture details in both front and back view renderings, maintaining 3D coherence and ensuring high alignment with the garment in the reference image.

We provide additional generated results guided by diverse reference images in A.3, showcasing the robustness of our method. Although the model is trained solely on T-pose character–garment pairs, it generalizes well to a wide range of scenarios involving different poses, viewpoints, and backgrounds, as well as real-character images. Specifically, we demonstrate its effectiveness in the following cases: **Half-Body Scenario**, **Body Occlusion Scenario**, and **Person Image Scenario**. Furthermore, we evaluate our method on GarmentCodeData Korosteleva et al. (2024) ( A.4), which further confirms its generalization ability across diverse garment meshes.

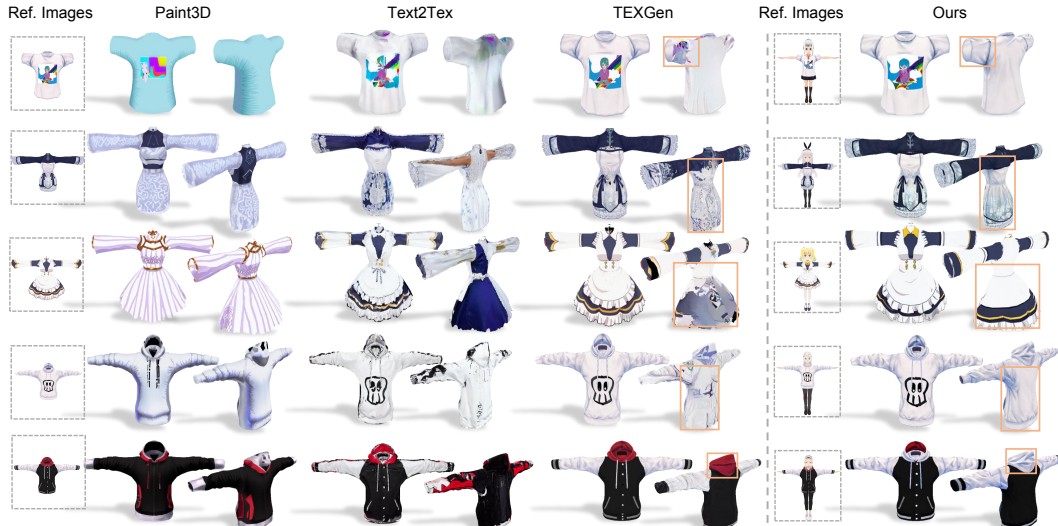

Figure 5: Qualitative comparison with SOTA methods. Our approach produces garment textures that closely match the reference images, even when characters are present. Compared to other methods, the results are more detailed and maintain better 3D consistency. Please zoom in to view details.

| Methods | FID↓ | KID($\times 10^{-3}$)↓ | Runtime↓ |
|---|---|---|---|
| TEXTure Richardson et al. (2023) | 75.33 | 17.79 | ∼ 140s |
| Text2Tex Chen et al. (2023a) | 77.28 | 13.39 | ∼ 940s |
| Paint3D Zeng et al. (2024) | 71.32 | 7.35 | ∼ 220s |
| TEXGen Yu et al. (2024) | 46.90 | 3.37 | ∼ 10s |
| TEXGen† Yu et al. (2024) | 44.82 | 2.96 | ∼ 10s |
| Ours | **39.79** | **1.01** | ∼ **4s** |

Table 1: Quantitative comparison against state-of-the-art methods. † denotes that TexGen is trained on our dataset. We report the FID, KID and runtime. Our method achieves the best performance.

**Quantitative comparisons.** Table 1 reports a quantitative comparison with prior garment texture synthesis methods. For FID Heusel et al. (2017) and KID Bińkowski et al. (2018), we render $1024 \times 1024$ images of each mesh with synthesized textures from 20 fixed viewpoints. Runtime is measured as the time to generate a full texture map for a given reference image and mesh. Our method achieves the best FID and KID scores, surpassing TEX-Gen Yu et al. (2024) by 7.1 (FID) and 2.2 (KID), showing clear gains in fidelity. For runtime, all methods are tested on a single NVIDIA A100 GPU. Our model generates a UV texture map in only 4 seconds, significantly faster than others, thanks to its lightweight yet effective design.

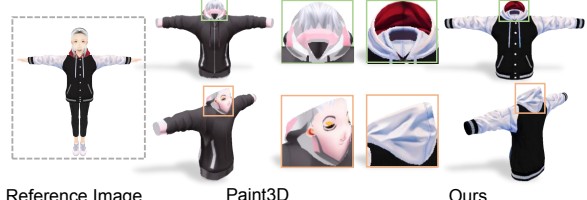

Figure 6: Comparison with Paint3D Zeng et al. (2024) using a character image as reference.

**Character image as reference.**

Paint3D Zeng et al. (2024) supports character images as input. As shown in Figure 6, we use a character image as a reference to visually compare our method with Paint3D Zeng et al. (2024). Paint3D struggles to accurately distinguish garment types in the reference image, often incorporating facial features into the generated texture, resulting in poor-quality outputs. In contrast, our method precisely identifies the garment regions in the reference image, ensuring the generated textures are well-aligned with the reference garment. Additionally, our approach avoids introducing irrelevant content, maintaining high texture fidelity throughout the generation process.

## 5.2 ABLATION STUDY

We first investigate the two key components, *i.e.* UV position map and type selection module. The experimental results are summarized in Table 2.

| Methods | FID↓ | KID↓ |
|---|---|---|
| *w/o* type select module | 45.08 | 1.30 |
| *w/o* uv position map | 47.82 | 1.45 |
| Ours | 39.79 | 1.01 |

Table 2: Evaluation of modules in our method. This demonstrates the effectiveness of each component. Our full model achieves optimal performance.

Reference Image    w/o type selection    w/ type selection

Figure 7: The visual comparison with and without type selection module.

**The effectiveness of UV position map.** We remove the UV position map from the U-Net's input, leaving only the mask image to provide basic shape information for texture generation. This modified model achieves FID and KID scores of $47.82$ and $1.45$, respectively. The performance are significantly worse than our full model. These results demonstrate the effectiveness of the UV position map, which plays a crucial role in providing structural guidance for texture generation.

**The effectiveness of Type Selection Module.** To evaluate the role of the type selection module, we remove it from the framework, leaving the model without explicit garment-type guidance. This variant yields substantially worse results, with FID and KID scores of $45.08$ and $1.30$, respectively. As shown in Figure 7, the absence of type signals causes the model to misplace textures across garment regions (*e.g.*, transferring skirt details onto a shirt). In contrast, our full model leverages type embeddings to enforce correct texture-to-garment alignment, producing more accurate and coherent outputs. Both quantitative and qualitative comparisons confirm the effectiveness of the type selection module.

We further analyze the robustness of our method under two challenging conditions: (1) when the reference image and UV map are misaligned, and (2) when the reference image does not contain any character instances.

**Reference image - UV map mismatch.** In this challenging case, the reference image depicts a dress while the UV map corresponds to a 3D T-shirt. Despite this mismatch, our method generates coherent textures, as detailed in A.5.

**Reference image without characters.** We compare the results of our method using reference images both with and without characters. In both cases, the method performs well, as shown in A.6.

**Dynamic results.** To further validate our approach, we present a dynamic example in A.7. This example demonstrates that our method not only maintains high-quality texture generation under static settings but also preserves consistency and robustness across temporal variations, highlighting its effectiveness in more practical and challenging scenarios.

## 6 CONCLUSION

In this work, we introduce GarmentPainter, a novel framework for high-quality, 3D-consistent garment texture generation that prioritizes efficiency. Our approach addresses the key challenges found in existing methods, including 3D consistency, garment-aware generation, and speed. By conditioning a diffusion model on both a character reference image and a UV position map, we ensure accurate texture alignment while preserving garment structure. Furthermore, our type selection module provides fine-grained control over specific garment components, effectively preventing texture artifacts. Extensive experiments show that GarmentPainter outperforms state-of-the-art methods, offering an effective and scalable solution for realistic 3D garment visualization.

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

# A APPENDIX

## A.1 RASTERIZATION

The shapes of garment meshes in the VRoid data are also defined by an alpha channel within the UV texture. Before rendering our images, we rasterize the UV position map using the alpha channel to isolate the mesh positions. We compute the mesh's scale and translation with these selected mesh positions. We then translate the mesh to align with the origin of the coordinate system and scale it to fit within the $[-1, 1]$ range. This process ensures that the rendered portion of each mesh remains centered in the final image. After translating the mesh, we re-rasterize the UV position map to align it within the $[-1, 1]$ range. For the UV mask image.

## A.2 PRESETTING.

State-of-the-art methods require different types of inputs. To ensure a fair comparison, we carefully configure the inputs for each approach as follows:

- **Text2Tex** Chen et al. (2023a) and **TEXTure** Oechsle et al. (2019) originally rely on text prompts. To integrate our reference image, we follow TexGen Yu et al. (2024) by substituting the first view of the diffusion output with the rendered view from the garment ground-truth UV texture. Since these methods require a text prompt, we provide one generated by GPT Achiam et al. (2023).

- **Paint3D** Zeng et al. (2024) uses IP-Adapter Ye et al. (2023) for image-conditioned generation. To remain consistent with the original paper, we render the view image to match the input format used by TEXTure Oechsle et al. (2019). We also provide a text prompt. Notably, we render only the garment mesh, excluding any human-related parts, making the task easier compared to using a full character image.

- **TEXGen** Yu et al. (2024) requires a reference image that exactly matches a specific garment-mesh view, along with the view pose. This allows projecting the view back onto the mesh to obtain an incomplete UV texture, which TEXGen Yu et al. (2024) then completes. We also provide text prompts for this method.

- **GarmentPainter** naturally supports character images and facilitates type-specific garment generation. To showcase our model's strengths, we directly use the character image as input. Since our approach bypasses cross-attention modules, we do not use text prompts in our experiments.

All qualitative and quantitative experiments follow the above settings to ensure a fair comparison.

## A.3 MORE RESULTS OF OUR METHODS.

In Figure 8, we showcase additional results demonstrating the robustness of our method. Despite being trained exclusively on T-pose character-garment image pairs, it generalizes effectively to various scenarios, underscoring its efficacy in garment texture generation: 1) **Half-Body Scenario** Even if the reference image depicts only the upper body, our method accurately captures the garment's texture details. 2) **Body Occlusion Scenario** When the reference garment is partially obscured, a common real-world challenge, our method still faithfully extracts texture details. 3) **Person Image Scenario** Although our training set includes no real human images, our approach can generate high-quality textures from such references.

Moreover, our method maintains consistent quality across front and back views, highlighting its reliability. Notably, even when the reference garment's shape differs from the mesh, our method produces coherent and visually appealing results.

## A.4 EVALUATION ON GARMENTCODEDATA

To further demonstrate the generalization ability of our method across diverse garment meshes, we conduct evaluations on GarmentCodeData Korosteleva et al. (2024) which is a large-scale synthetic

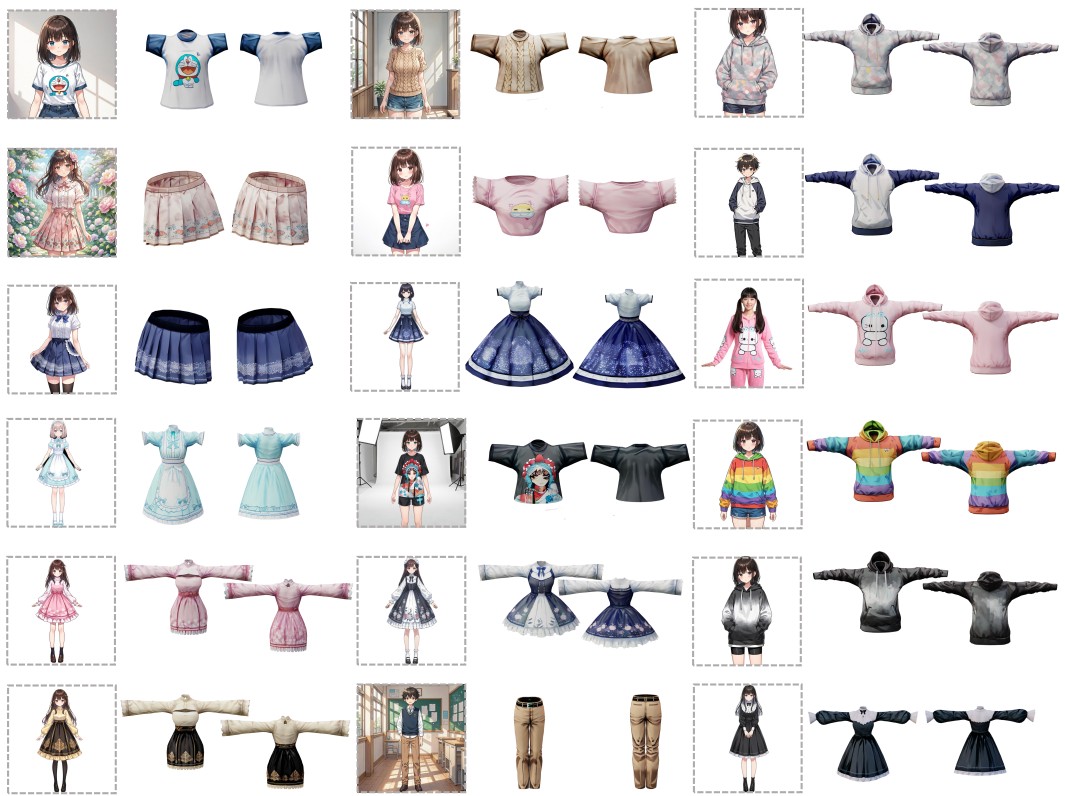

Figure 8: Additional results of our method. We showcase a variety of reference images, including cartoon characters, real people, and scenes with complex backgrounds. Across these diverse inputs, our method consistently generates high-quality texture maps that accurately reflect the garment depicted in each reference image.

dataset of 3D made-to-measure garments with sewing patterns. This dataset provides a wide variety of garment categories and mesh structures, posing greater challenges for consistent texture generation. As shown in Figure 9, our method produces high-quality and semantically aligned textures across different garment types, thereby confirming its robustness and effectiveness in handling complex and diverse mesh representations.

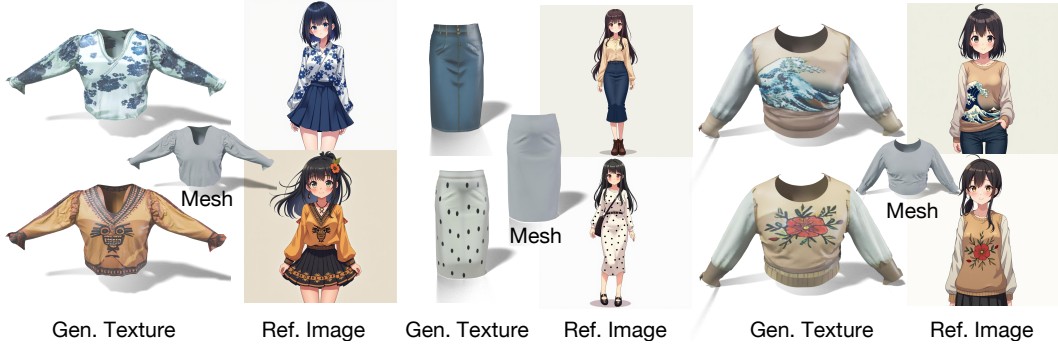

Figure 9: Evaluation on GarmentCodeData Korosteleva et al. (2024) demonstrates the strong generalization ability of our method across diverse garment meshes.

## A.5 REFERENCE IMAGE - UV MAP MISMATCH

Figure 10 illustrates a particularly challenging case in which the reference image depicts a dress while the UV map corresponds to a 3D T-shirt. Such a mismatch in garment category and geometry often leads to severe texture distortion or unrealistic mapping in baseline methods. In contrast, our method is able to generate visually coherent and semantically consistent textures that adapt naturally to the target UV map. This result highlights not only the robustness of our framework to cross-category discrepancies but also its strong generalization ability to garment types unseen during training, further demonstrating the versatility of our approach in real-world applications.

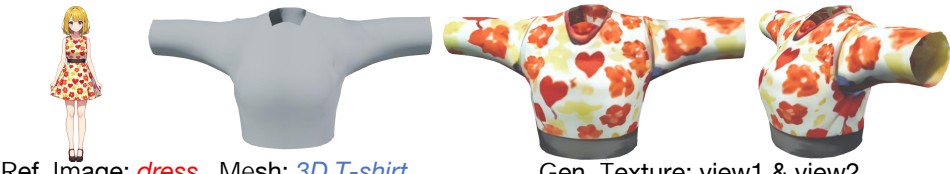

Ref. Image: *dress*   Mesh: *3D T-shirt*          Gen. Texture: view1 & view2

Figure 10: Evaluation under challenging setting: the reference image and UV map are mismatched.

## A.6 REFERENCE IMAGE WITHOUT CHARACTERS

Table 3 compares the performance of our method using reference images both with and without characters. When character information is present, the model generates realistic and semantically consistent outputs. In the absence of characters, it still captures the essential style and appearance from the garment region alone, producing coherent and visually plausible textures. These results confirm that our approach is not reliant on character cues and remains robust across both settings.

| Methods | FID↓ | KID($\times 10^{-3}$)↓ |
|---------|------|------------------------|
| *w/o* character | 38.54 | 0.98 |
| *w/* character | 39.79 | 1.01 |

Table 3: Comparison of our method using reference images with and without characters, showing robust performance and visually consistent textures in both cases.

## A.7 DYNAMIC RESULTS

To further substantiate the effectiveness of our approach, we provide a dynamic example in Figure 11. Unlike static demonstrations, which primarily highlight visual fidelity on individual frames, this dynamic case emphasizes the temporal stability and adaptability of our method. The results show that our approach is capable of maintaining high-quality texture generation across continuous frame sequences, thereby ensuring that garment details remain consistent under motion and viewpoint changes. Moreover, even when subjected to temporal variations that often cause flickering or texture distortion in baseline methods, our model exhibits robust performance with smooth and coherent outputs. This observation highlights not only the accuracy of the generated textures in isolated frames but also the reliability of our method in more practical, real-world scenarios where garments and characters are inherently dynamic.

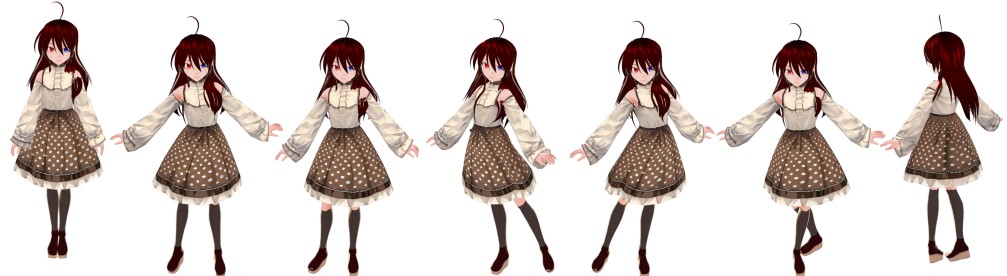

Figure 11: Dynamic results. We present a dynamic example that demonstrates the effectiveness and robustness of our method.

