# OpenReview forum: "GarmentPainter: Efficient 3D Garment Texture Synthesis with Character-Guided Diffusion Model"
_ICLR.cc/2026/Conference — Submitted to ICLR 2026_

### Official Review · Reviewer_4FyR · 2025-10-28

**Soundness:** 3
**Presentation:** 2
**Contribution:** 2
**Rating:** 4
**Confidence:** 5

**Summary:**

This paper presents GarmentPainter, a framework for synthesizing garment textures in UV space. It leverages a UV position map as a 3D structural guide to ensure texture consistency across the garment surface. Experimental results show that GarmentPainter achieves state-of-the-art performance in terms of visual fidelity, 3D consistency, and computational efficiency.

However, the overall framework lacks substantial novelty, as employing a UV position map for 3D structural guidance is an established practice. In addition, the motivation for introducing the type selection module is not clearly justified.

**Strengths:**

1. An application-driven study that delivers convincing results.
2. Construct a high-quality garment dataset tailored for texture generation.

**Weaknesses:**

1. The paper shows limited novelty. Using a UV position map is not a new idea, and Paint3D also support direct generation in UV space.

2. The introduction of the type selection module is not well-motivated. Why not directly use the cloth type as a text prompt instead of introducing an additional type encoder? It would be helpful to report the performance when using cloth type as a textual condition. Moreover, the claim that the type selection module works without alignment between the reference image and the 3D mesh seems somewhat overstated.

3. The discussion of 2D virtual try-on in the related work section appears unnecessary, since the paper’s focus is on 3D texture generation.

4. The discussion of limitations is insufficient. How does the proposed method perform on fine or automatically generated UV maps, such as those produced by atlas-based methods?

5. Minor issue: There are a few typos in Figure 3.

**Questions:**

1. The paper shows limited novelty. Using a UV position map is not a new idea, and Paint3D also support direct generation in UV space.

2. The introduction of the type selection module is not well-motivated. Why not directly use the cloth type as a text prompt instead of introducing an additional type encoder? It would be helpful to report the performance when using cloth type as a textual condition. Moreover, the claim that the type selection module works without alignment between the reference image and the 3D mesh seems somewhat overstated.

3. The discussion of limitations is insufficient. How does the proposed method perform on fine or automatically generated UV maps, such as those produced by atlas-based methods?

---

> ### Author Response · Authors · 2025-11-22
>
> Thanks for your insightful feedback and your time in reading our paper. We respond to each of your comments one by one.
>
> **Weakness 1 & Question 1:** The paper shows limited novelty. Using a UV position map is not a new idea, and Paint3D also support direct generation in UV space.
>
> **Answer :** We would like to clarify that our novelty does not lie in simply adopting the UV position map, but rather in how we integrate UV positional information and reference-image conditions into a unified, lightweight conditioning mechanism. This design is fundamentally different from both Paint3D and TEXGen.
>
> **Novelty of Our Method: Unified Conditioning Strategy.** Although both Paint3D and TEXGen utilize UV position maps, their conditioning mechanisms are substantially different from ours:
>
> **Paint3D** injects positional signals through an additional position-encoder adapter added to the backbone.
>
> **TexGen** employs a 2D–3D hybrid architecture to inject UV positional information into the generation process, which makes it difficult to leverage pretrained diffusion models. As a result, it trains the entire generation model from scratch.
>
> In contrast, our method introduces a unified and minimal conditioning strategy. We concatenate the UV position map with the generated texture along the channel dimension, leveraging the natural spatial alignment between the UV texture and the UV position map. We further concatenate the reference image along the spatial dimension, enabling the original diffusion model’s attention mechanism to effectively exchange information. Benefiting from this design, the only architectural modification required is a simple adjustment to the input convolution layer to accommodate the expanded input dimensionality. Through this unified conditioning, both texture cues from the reference image and structural cues from the UV position map are jointly propagated, allowing the generation process to be effectively guided by texture information while preserving UV-structural consistency.
>
> More details are provided in Sec.4.1/4.2, and the qualitative as well as quantitative results further validate the advantages of our method.
>
> ---
>
> **Weakness 2 & Question 2:** The introduction of the type selection module is not well-motivated. Why not directly use the cloth type as a text prompt instead of introducing an additional type encoder? It would be helpful to report the performance when using cloth type as a textual condition. Moreover, the claim that the type selection module works without alignment between the reference image and the 3D mesh seems somewhat overstated.
>
> **Answer :** Thank you for the insightful comments. We will clarify the motivation and provide the requested comparison.
>
> **(1) Why not directly use cloth type as a text prompt?**
> Our current pipeline removes text cross-attention because the conditioning is injected through latent-space feature modulation from the reference image. Adding textual prompts would re-introduce cross-attention blocks, increasing both memory usage and inference latency. To address this concern, we have conducted an additional experiment that uses the cloth type as a textual condition. The results show comparable quality, but the text-prompt variant is $\sim 2.5$ slower due to the cross-attention computation. We will include these quantitative results in the revised version.
>
> Comparison between text cross-attention and type-selection module:
>
> | Methods               | FID↓  | KID↓  | Runtime↓ |
> |-----------------------|-------|-------|----------|
> | text cross attention  | 41.02 | 1.14  | ~10s     |
> | type select module    | 39.79 | 1.01  | ~4s      |
>
> **(2) Motivation of the type-selection module.**
> The type-selection encoder is a lightweight alternative that avoids cross-attention while providing explicit control over garment type, leading to more stable conditioning in our ablation.
>
> **(3) Claim about not requiring alignment between the reference image and the 3D mesh.**
> In the revision, we will soften this claim and clarify that the method is robust to imperfect alignment, rather than completely independent of alignment.
>
> ---

---

> > ### Author Response · Authors · 2025-11-22
> >
> > **Weakness 3:** The discussion of 2D virtual try-on in the related work section appears unnecessary, since the paper’s focus is on 3D texture generation.
> >
> > **Answer :** Thank you for the suggestion. We agree that the 2D virtual try-on discussion is unnecessary, and we will remove the irrelevant details in the revised version to keep the related work focused on 3D texture generation.
> >
> > ---
> >
> > **Weakness 4 & Question 3:** The discussion of limitations is insufficient. How does the proposed method perform on fine or automatically generated UV maps, such as those produced by atlas-based methods?
> >
> > **Answer :** Thank you for pointing this out. In the revision, we will expand the limitation section. Our method is designed for structured UV layouts with consistent semantic regions. When applied to fine-grained or automatically generated UV maps (e.g., atlas-based unwrapping), the performance degrades due to fragmented islands and the loss of part-level continuity. We will include qualitative examples and clarify that handling arbitrary atlas UVs remains a limitation of our current pipeline.
> >
> > ---

---

> ### Comment · Reviewer_4FyR · 2025-11-27
>
> Thank you for the authors’ response. After carefully reading the rebuttal and the other reviewers’ comments, I have decided to give  a **rejection** recommendation. My justifications are as follows:
> - Regarding the unified conditioning strategy, the authors claim that it leverages the natural spatial alignment between the UV texture and the UV position map. However, **the core contribution of concatenating the UV position map with the generated texture along the channel dimension of appears rather trivial from my perspective.**
> - The introduction of the type-selection module is not sufficiently motivated. Although the type-selection encoder offers a lightweight alternative to cross-attention, it sacrifices the diversity and generative flexibility enabled by text prompts, effectively reducing the model’s capability to three predefined tasks.

---

> ### Author Response · Authors · 2025-12-01
>
> Thank you for the additional feedback. We respectfully address the two concerns as follows.
> 1. On the unified conditioning strategy。
> We understand the reviewer’s impression that concatenating the UV position map may appear simple at first glance. However, the contribution is not the operation itself, but the task-specific inductive bias it injects into the diffusion pipeline.
> Unlike natural images, UV layouts contain discontinuities, fragmented islands, and non-Euclidean topology, making texture generation in UV space inherently difficult. Existing diffusion backbones are not designed for this domain.
> Our conditioning strategy enables the network to explicitly perceive the semantic UV coordinates at every pixel, effectively restoring spatial continuity in an otherwise discontinuous domain. This greatly stabilizes training and improves cross-view consistency, as demonstrated across our experiments (including reference–UV mismatch and out-of-domain evaluations).
> Thus, while the mechanism is lightweight, it is not trivial—it is a deliberate design tailored to the unique structure of UV space, and experimentally shown to be both effective and efficient.
>
> 2. On the type-selection module and “loss of generative flexibility”
> We also respectfully clarify a misunderstanding. In garment texture generation, the source of generative diversity is the reference image—not the text prompt. Reference images contain far richer and more precise visual informationthan short textual descriptions. Removing text cross-attention does not reduce the diversity or flexibility of the generated textures. The purpose of the type-selection module is not to constrain the generative space, but to provide lightweight and explicit structural control (top/bottom/one-piece). It replaces a heavy cross-attention mechanism with a compact encoder: reducing computation and parameters, avoiding the ambiguity of text descriptions, and improving stability during training. The richness of generated textures still comes entirely from the reference image. The module only determines the garment category—it does not limit or predefine the texture variations.

---

### Official Review · Reviewer_4r63 · 2025-10-29

**Soundness:** 2
**Presentation:** 2
**Contribution:** 2
**Rating:** 2
**Confidence:** 4

**Summary:**

This paper proposes **GarmentPainter**, a diffusion-based framework for generating 3D garment textures directly in UV space. The method modifies Stable Diffusion 1.5 by removing text cross-attention and injecting multiple VAE-encoded latent modalities (reference image, UV position map, UV texture/mask) to guide generation. A small type-selection module is added to control generation across top/bottom/one-piece garments.

**Strengths:**

- **Data Contribution**: The authors curate a garment-specific dataset with UV maps, reference images, and mask/position data, which is valuable for this niche area of 3D garment texturing.
- **Structural Innovation on SD1.5**: The way the authors adapt SD1.5 — particularly replacing text cross-attention with multi-modal VAE latent conditioning — is a novel and neat architectural modification that simplifies conditioning without heavy architectural changes.
- **Experimental Soundness**: The ablation studies are well-designed and convincingly demonstrate the necessity of each component (UV position map, type selection), showing clear performance degradation when removed.

**Weaknesses:**

**Concerns on Generalization**
> *[Sec.3 L206-209]* “Ultimately, we curate a dataset comprising 7,579 clothing items, including 3,703 tops, 2,114 bottoms, and 1,762 one-piece garments.”
- Although the dataset creation is commendable, the total scale (~7.6k) appears small relative to the architectural modifications made to SD1.5 (multi-modal VAE inputs, removal of text cross-attention). It raises the question of whether such a limited dataset is sufficient to grant **true generalization** rather than overfitting.
- The paper does not specify the **train/validation/test split**, nor clarify whether evaluation is on the same data source or on external data. This omission further amplifies concerns regarding generalization.
- UV robustness is under-explored. All data are processed in Blender with a consistent UV unwrapping workflow. It remains unclear whether the method can generalize to **other UV layouts**, especially auto-generated or platform-specific UVs, which often introduce discontinuities.

**Limited Novelty**
- While the architectural attempt is appreciated, using a UV coordinate/position map to maintain spatial continuity is not new — both **Paint3D** and **TEXGen** adopt similar strategies.
- Furthermore, directly generating UV maps inherently struggles to guarantee 3D spatial consistency due to UV seam discontinuities and varying unwrapping conventions. This is why many recent works still rely on multi-view synthesis[1,2,3] for texture painting. That said, for **garment textures**, where UVs tend to be more regular, the approach is acceptable and practical.

**Relatively Outdated Foundation**
- From a generative model standpoint, the field has rapidly evolved beyond SD1.5 (e.g., SDXL, FLUX, Qwen-Image), with substantially improved image quality, resolution, and visual priors. Operating on SD1.5 limits output resolution to 512px and inherently lags behind current capabilities.
- The paper neither discusses nor compares against more recent texture-oriented or multi-view-consistent generation approaches such as: Mv-Adapter[1], FlexiTex[3], which would provide a more up-to-date benchmark of competitiveness.

---

**Reference**

[1] "Mv-adapter: Multi-view consistent image generation made easy." *in ICCV 2025*.

[2] "Mvpaint: Synchronized multi-view diffusion for painting anything 3d." *in CVPR 2025*.

[3] "FlexiTex: Enhancing Texture Generation via Visual Guidance." *in AAAI 2025*.

**Questions:**

See weakness

---

> ### Author Response · Authors · 2025-11-22
>
> Thanks for your insightful feedback and your time in reading our paper. We respond to each of your comments one by one.
>
> **Weakness 1:** Although the dataset creation is commendable, the total scale (~7.6k) appears small relative to the architectural modifications made to SD1.5 (multi-modal VAE inputs, removal of text cross-attention). It raises the question of whether such a limited dataset is sufficient to grant true generalization rather than overfitting.
>
> **Answer :** Thank you for raising this important concern. Our design principle is to fully leverage the strong priors of pretrained diffusion models rather than training a texture generator from scratch. Consequently, only a relatively small amount of task-specific data is required to adapt the model to garment texture generation, and we do not observe overfitting in practice.
>
> To validate generalization, we evaluate our method on GarmentCodeData, where garments follow industry-standard UV layouts. Although our model is not trained on this dataset, it still produces coherent and high-quality textures, demonstrating strong generalization to unseen meshes and UV configurations. Please refer to Appendix A.4 for more details.
>
> We also test our method with both AI-generated images and real photographs as references. AI-generated images contain complex patterns, while real-person photos involve diverse and challenging poses. Across these settings, our method consistently generates high-quality textures, further confirming its robustness. Please refer to Appendix A.3 for more details.
>
> Moreover, Appendix A.5 presents additional experiments under reference–UV mismatch, where the reference image and UV layout have intentional inconsistencies. Our method still produces reasonable results under these challenging conditions, further supporting its strong generalization capability.
>
> ---
>
> **Weakness 2:** The paper does not specify the train/validation/test split, nor clarify whether evaluation is on the same data source or on external data. This omission further amplifies concerns regarding generalization.
>
> **Answer :** We apologize for not clearly stating our dataset split earlier. We use 150 samples out of 7579 as the test set, including 50 tops, 50 bottoms, and 50 one-piece garments.
>
> ---
>
> **Weakness 3:** UV robustness is under-explored. All data are processed in Blender with a consistent UV unwrapping workflow. It remains unclear whether the method can generalize to other UV layouts, especially auto-generated or platform-specific UVs, which often introduce discontinuities.
>
> **Answer :** This concern arises from a misunderstanding. The VRoid dataset already provides high-quality UV parameterization, and therefore we do not use Blender for UV unwrapping at any stage. Our method operates directly on the original UV layouts from VRoid, which are widely used in 3D character and garment creation pipelines.  Our task focuses on garment textures, and garment typically exhibits structured and semantically coherent geometry, making it suitable for regular UV layouts used in garment texture generation. This is fundamentally different from general-purpose unwrapping of arbitrary 3D shapes, where highly irregular UV discontinuities are common.  To further assess generalization beyond VRoid UVs, we also evaluate our method on the GarmentCodeData dataset, which contains industry-standard UV parametrization created by professional garment design tools. Our method performs consistently well on this dataset, demonstrating that the proposed  framework is robust across diverse real-world garment UV parameterizations.
>
> ---

---

> ### Author Response · Authors · 2025-11-22
>
> **Weakness 4:** While the architectural attempt is appreciated, using a UV coordinate/position map to maintain spatial continuity is not new — both Paint3D and TEXGen adopt similar strategies.
>
> **Answer :** We would like to clarify that our novelty does not lie in simply adopting the UV position map, but rather in how we integrate UV positional information and reference-image conditions into a unified, lightweight conditioning mechanism. This design is fundamentally different from both Paint3D and TEXGen.
>
> **(1). Novelty of Our Method: Unified Conditioning Strategy.**
>
> Although both Paint3D and TEXGen utilize UV position maps, their conditioning mechanisms are substantially different from ours:
>
> **Paint3D:** injects UV position signals through an additional position-encoder adapter added to the backbone.
>
> **TEXGen:** employs a 2D–3D hybrid architecture to inject UV positional information into the generation process, which makes it difficult to leverage pretrained diffusion models. As a result, it trains the entire generation model from scratch.
>
> In contrast, our method introduces a unified and minimal conditioning strategy.  We concatenate the UV position map with the generated texture along the channel dimension, leveraging the natural spatial alignment between the UV texture and the UV position map.  We further concatenate the reference image along the spatial dimension, enabling the original diffusion model’s attention mechanism to effectively exchange information. Benefiting from this design, the only architectural modification required is a simple adjustment to the input convolution layer to accommodate the expanded input dimensionality.  Through this unified conditioning, both texture cues from the reference image and structural cues from the UV position map are jointly propagated, allowing the generation process to be effectively guided by texture information while preserving UV-structural consistency.
>
> More details are provided in Sec.4.1/4.2, and the qualitative as well as quantitative results further validate the advantages of our method.
>
> **(2). Novelty: Content-Aware Generation Without Strict Geometry Alignment.**
>
> Unlike prior methods that require tightly geometry-aligned inputs, otherwise often producing mismatched or incorrect content (as shown in Fig.6) our method achieves content-aware generation through:the type-selection module, and our data construction strategy.  Together, these components allow the diffusion model to reason about semantic garment context, rather than relying solely on geometric correspondence. As illustrated in Fig. 4, the attention maps demonstrate that our method is capable of context-aware generation.  This capability is absent in both Paint3D and TEXGen.
>
> ---
>
>
>
> **Weakness 5:** Furthermore, directly generating UV maps inherently struggles to guarantee 3D spatial consistency due to UV seam discontinuities and varying unwrapping conventions. This is why many recent works still rely on multi-view synthesis[1,2,3] for texture painting. That said, for garment textures, where UVs tend to be more regular, the approach is acceptable and practical.
>
> **Answer :** We believe that the strength of multi-view methods largely comes from leveraging pretrained diffusion models, which can synthesize high-quality and diverse viewpoints. However, multi-view pipelines still suffer from fundamental limitations. In contrast, directly applying pretrained diffusion models to UV-texture generation offers a simple and effective way to address these issues.
>
> Multi-view synthesis is inherently prone to view-dependent inconsistencies—such as perspective distortion, cross-view mismatches, and projection errors—which often lead to visible seams and misaligned patterns. More importantly, occluded regions remain ambiguous, even in state-of-the-art commercial systems, because no multi-view setup can provide full surface coverage.
>
> Direct UV-map generation avoids these problems altogether. In UV space, each surface point is represented once, eliminating ambiguities caused by viewpoint differences. The only discontinuities come from UV-island boundaries, which are classical unwrapping artifacts and can be placed in visually insignificant regions using modern UV tools.
>
> As the reviewer noted, our method acceptable under regular UV layouts. Garment meshes naturally exhibit structured UV maps, making our approach highly suitable for clothing textures. Our multi-view rendering in Fig. 4 also confirms that the generated textures are spatially coherent and free from multi-view artifacts.
>
> We further note that recent advances in semantic-aware UV unwrapping encourage more regular and consistent layouts, suggesting that UV-space generation may generalize well beyond garments type as UV techniques continue to improve.
>
> ---

---

> ### Author Response · Authors · 2025-11-22
>
> **Weakness 6:** From a generative model standpoint, the field has rapidly evolved beyond SD1.5 (e.g., SDXL, FLUX, Qwen-Image), with substantially improved image quality, resolution, and visual priors. Operating on SD1.5 limits output resolution to 512px and inherently lags behind current capabilities.
>
> **Answer :** Thank you for the suggestion. We implement our method on top of the Flux-Fill inpainting backbone, and the corresponding experimental results are provided in the below table. Due to GPU memory limitations, we adopt the same training resolution as SD-V1.5.
>
> Backbone comparison:
>
> | Methods                    | FID↓  | KID↓  | Runtime↓ |
> |---------------------------|-------|-------|----------|
> | Flux-Fill backbone        | 39.31 | 1.0   | ~30s     |
> | SD-V1.5 backbone(Ours)    | 39.79 | 1.01  | ~4s      |
>
> Our framework is backbone-agnostic, and we expect that integrating more advanced inpainting models (e.g., Flux-Kontext) will further improve performance. We will clarify this extensibility in the revision.
>
> ---
>
> **Weakness 7:** The paper neither discusses nor compares against more recent texture-oriented or multi-view-consistent generation approaches such as: Mv-Adapter, FlexiTex, which would provide a more up-to-date benchmark of competitiveness.
>
> **Answer :** Thank you for the suggestion. Following your recommendation, we now include comparisons with recent methods, including MV-Adapter (ICCV-2025) and MV-paint (CVPR-2025). As Flexible Texture is not open-sourced, our comparison is conducted against MVAdapter and MVPaint. The updated table shows that our method consistently achieves higher scores across all metrics, further demonstrating its advantages in garment texture generation. We will add the comparisons in the revision.
>
> Table 2. Comparison with recent methods:
>
> | Methods               | FID↓  | KID↓ | Runtime↓ |
> |-----------------------|-------|------|----------|
> | MVPaint               | 50.94 | 4.32 | ~97s     |
> | MVAdapter             | 42.76 | 1.63 | ~33s     |
> | Ours   | 39.79 | 1.01 | ~4s      |

---

> > ### Comment · Reviewer_4r63 · 2025-11-27
> > **Follow up discussion on author's response**
> >
> > Thank you for the authors’ response. I have carefully read the rebuttal and appreciate the additional experiments, which demonstrate the effectiveness of the proposed method under this specific setting. However, I still have some concerns:
> >
> > **Generalization**. As is well known, widely used controllable image generation methods such as ControlNet typically require training data that is `robust across both small (<50k) and large (>1M) scales`. Moreover, UV map generation differs significantly in layout from natural images produced by traditional diffusion models; discontinuity in UV space is notoriously difficult to overcome. I remain doubtful about the generalization ability given the `~7k scale of the dataset` used in this paper. To my understanding, the authors also **do not perform augmentation based on different UV unwrapping** strategies—for instance, using multiple unwrapping methods and baking textures accordingly. As far as I know, if the authors carefully cleaned the Objaverse dataset, it should be possible to filter out a considerable number of suitable training samples.
> >
> > What I would like to see is: given the same mesh and input image pair, results under **different UV unwrapping strategies (e.g., xatlas)**. If the model is not robust to automatic unwrapping strategies such as xatlas, then the proposed colorization pipeline would only work for artist-designed meshes and would not generalize to AI-generated clothing meshes—greatly limiting the practical utility of the paper.
> >
> > Additionally, on the side of input images, the authors always render the front and back view of the model in a very fixed manner. In most paper's showcases, AI-generated images also use relatively standard model poses. Because the conditioning is so uniform, this further raises concerns about robustness to input variations. For example, when the character has hands in front of the chest, or when the input is a side view—cases involving **strong occlusions and invisible parts**—can the model reasonably generate the missing textures?
> >
> > In summary, I believe that the overall completeness of this work is, frankly, in a relatively rough state. The issues I raised reflect this roughness across multiple aspects. Overall, I still hold negative opinion toward the current version of the work.

---

> ### Author Response · Authors · 2025-12-01
>
> Thank you for the follow-up comments. We address the reviewer’s concerns with a concise and academically focused clarification.
> 1. Generalization and dataset scale
> Our task differs fundamentally from general controllable image generation (e.g., ControlNet). Those models learn new control pathways and must acquire broad visual semantics from scratch, which demands large datasets.
> In contrast, our method adapts a pretrained diffusion prior to the UV domain, and the task relies heavily on garment-specific structural regularities (e.g., consistent part topology, fabric arrangement). As shown in recent works such as Flux-Kontext, small-scale finetuning (<5k samples) is sufficient when adaptation leverages a strong pretrained backbone.
> We provide two forms of empirical evidence supporting generalization:
> (a) Reference-image generalization
> The method produces stable textures on real photographs and AI-generated inputs, which exhibit diverse materials, colors, and poses. These results (manuscript + supplementary) demonstrate robustness to appearance variations beyond the training domain. (b) Mesh/UV generalization
> The method transfers successfully to GarmentCodeData meshes with entirely different UV layouts, without any retraining. This indicates that the proposed conditioning mechanism is not tied to VRoid UV templates and is compatible with other structured UV parameterizations.
> These two axes jointly support the method’s capacity to generalize despite a ~7k dataset.
>
> 2. On UV unwrapping strategies (e.g., xatlas)
> Automatic UV unwrapping algorithms (e.g., xatlas) produce UV maps with: fragmented semantic islands, inconsistent chart adjacency, loss of garment-level topology.
> These structural properties fundamentally differ from apparel-oriented UV layouts used in industry and in most creator-oriented pipelines. The proposed method is targeted at these structured UV parameterizations, which remain the dominant representation for clothing assets, including those generated by modern AI-based 3D garment models.
> Evaluating the method under xatlas, which destroys such structure, is not aligned with the intended application domain of garment texture generation. We will make this task-specific assumption more explicit.
>
> 3. Robustness to input-view variations
> Beyond the canonical front/back renderings, our evaluation includes: real images with body-pose variability and partial self-occlusions, AI-generated references containing complex and non-standard poses.
> The model produces coherent textures in these settings, and the results provided in the supplementary demonstrate robustness to moderately occluded or non-frontal views.

---

### Official Review · Reviewer_GpsL · 2025-10-29

**Soundness:** 3
**Presentation:** 3
**Contribution:** 3
**Rating:** 6
**Confidence:** 3

**Summary:**

The paper targets the intesting problem of 3D garment texture synthesis which is of great importance to the industry applications like games and animations. The main idea of the proposed GarmentPainter is to leverage the UV map to align 2D images to enable the 2D models to generate the 3D UV textures. To train the model, a new dataset is introduced with high-quality garments which is helpful for the garment texture generation problem. Experimental results validate the effectiveness and efficiency of the proposed algorithm.

**Strengths:**

* The problem of 3D garment generation is an important problem for industry applicatioons.
* The proposed garment dataset with high-quality garments should be useful to the 3D community.
* The proposed algorithm obtain promising results with sufficient ablation studies.

**Weaknesses:**

* More implementation details should be provided to facilitate the reproduction of the paper. Or it would be better to provide the code for reproduction.

* For the experimental results in Table 1, I would suggest to provide more comparisons against the papers published in the recent two years or in the year of 2025.

* In the experiments, the evaluation metric is based on FID and KID, which may not be consistent with human subjective evaluations. Thus, is it possible to provide a user study to verify the effectiveness of the proposed algorithm against the baselines.

* The algorithm is based on the inpainting model trained from SD v1.5. As there are rapid developement of the text-to-image community, how about the performance of the proposed algorithm if better inpainting models, like flux-kontext, is utilized?

**Questions:**

Please mainly address the questions in the weakness section. More specifically, the questions related with the experiments should be well addressed.

---

> ### Author Response · Authors · 2025-11-22
>
> Thanks for your insightful feedback and your time in reading our paper. We respond to each of your comments one by one.
>
> **Weakness 1:** More implementation details should be provided to facilitate the reproduction of the paper. Or it would be better to provide the code for reproduction.
>
> **Answer :**  We thank the reviewer for pointing this out. We use the Stable Diffusion v1.5 inpainting model as our texture-generation backbone. All UNet parameters except the cross-attention modules are finetuned with a constant learning rate of 1e-5 using AdamW. The model is trained for 70k steps on 7×A100 GPUs with a per-GPU batch size of 15. As detailed in Appendix A.1, we use 7,429 samples for training and 150 samples for testing (50 tops, 50 bottoms, 50 one-piece garments). We will include additional implementation details in the revised version to ensure reproducibility.
>
> ---
>
> **Weakness 2:** For the experimental results in Table 1, I would suggest providing more comparisons against papers published in the recent two years or in the year 2025.
>
> **Answer :** Thank you for the suggestion. Following your recommendation, we now include comparisons with recent methods, including MV-Adapter (ICCV 2025) and MV-Paint (CVPR 2025). The updated results show that our method consistently achieves higher scores across all metrics, further demonstrating its advantages in garment texture generation.
>
> #### Updated Results:
>
> | Methods              | FID ↓  | KID ↓  | Runtime ↓ |
> |----------------------|--------|--------|------------|
> | MVPaint              | 50.94  | 4.32   | ~97s       |
> | MVAdapter            | 42.76  | 1.63   | ~33s       |
> | Ours                 | 39.79  | 1.01   | ~4s        |
>
> We will add these comparisons in the revision.
>
> ---
>
> **Weakness 3:** The evaluation metric is based on FID and KID, which may not be consistent with human subjective evaluations. Is it possible to provide a user study?
>
> **Answer :** Thank you for the suggestion. We conducted a user study under the unified evaluation protocol. Following MVPaint’s evaluation criteria, all meshes are textured using both our method and the baselines.  Participants rated each result along two dimensions (1) seamlessness and (2) 3D consistency using a 1–5 scale. We collected scores from 5 participants and report the averaged results below.
>
> #### User Study Results:
>
> | Methods    | Seamless ↑ | 3D Consistency ↑ |
> |------------|-------------|------------------|
> | Paint3D    | 3.21        | 4.01             |
> | MVAdapter  | 3.87        | 4.62             |
> | MVPaint    | 3.71        | 4.05             |
> | TexGen     | 3.30        | 3.52             |
> | Ours       | 4.67        | 4.81             |
>
> ---
>
> **Weakness 4:** The algorithm is based on an inpainting model trained from SD v1.5. As there are rapid developments in the text-to-image community, how would the proposed algorithm perform if better inpainting models, like Flux-Kontext, were used?
>
> **Answer :** Thank you for the suggestion. We implemented our method on top of the Flux-Fill inpainting backbone, and the corresponding experimental results are shown below. Due to GPU memory limitations, we adopt the same training resolution as SD v1.5.
>
> #### Backbone Comparison:
>
> | Methods                 | FID ↓ | KID ↓ | Runtime ↓ |
> |-------------------------|--------|--------|-----------|
> | Flux-Fill backbone      | 39.31  | 1.00   | ~30s      |
> | SD-v1.5 backbone (Ours) | 39.79  | 1.01   | ~4s       |
>
> Our framework is backbone-agnostic, and we expect that integrating more advanced image-editing models (e.g., Flux-Kontext) will further improve performance. We will clarify this extensibility in the revision.

---

> > ### Comment · Reviewer_GpsL · 2025-11-26
> >
> > Thanks for the rebuttal. The rebuttal well  addressed most of my concerns.

---

### Official Review · Reviewer_ZtVi · 2025-11-03

**Soundness:** 3
**Presentation:** 3
**Contribution:** 3
**Rating:** 6
**Confidence:** 3

**Summary:**

The paper proposes GarmentPainter, a method that generates garment textures directly in UV space from a reference person image. The approach encodes the reference image and a UV position map into latents, concatenates them (with a masked UV texture channel) as UNet inputs, and injects a garment-type embedding (top/bottom/one-piece) into the diffusion timestep embedding for better control. The authors also describe a dataset of ~7.6k garments with reference images, UVs, type labels, and position maps, and report strong speed and competitive FID/KID against several baselines.

**Strengths:**

1. Simple and practical design: Minimal modifications to a standard inpainting diffusion backbone (channel concatenation + type embedding) make the method easy to implement and deploy.

2. Fast inference: Reported end-to-end UV generation is notably fast (single forward path), which is attractive for production pipelines compared with multi-view/iterative methods.

3. Workflow alignment: Accepting a person-in-context reference image maps well to real authoring scenarios, reducing pre- and post-processing overhead.

4. Clear data description: The paper gives a concrete breakdown of categories (top/bottom/one-piece), rendering protocol (front/back), and labeling procedure, which improves readability and reuse.

**Weaknesses:**

1. Fairness & reproducibility: Different baselines appear to be run under different input protocols (e.g., prompts, masks, background handling, illumination). This can bias comparisons. A single unified evaluation protocol (resolution, masking, prompts, backgrounds/lighting) and a reproducible package would strengthen claims.

2. 3D consistency metrics are thin: The evaluation focuses on image-space metrics (e.g., FID/KID) and runtime. It lacks direct measures of UV seam continuity, cross-view consistency, and geometric adjacency color differences, which are crucial for textures that must look coherent on a mesh.

3. Generalization beyond the training domain: The dataset leans toward a specific visual domain. Claims of robustness to real photos, complex materials, heavy patterns, and challenging poses would be more convincing with systematic out-of-domain tests and a failure-mode analysis.

**Questions:**

See weakness.

---

> ### Author Response · Authors · 2025-11-22
>
> Thanks for your insightful feedback and your time in reading our paper. We respond to each of your comments one by one.
>
> **Weakness 1: Fairness & reproducibility.**
> Different baselines appear to be run under different input protocols (e.g., prompts, masks, background handling, illumination). This can bias comparisons. A single unified evaluation protocol (resolution, masking, prompts, backgrounds/lighting) and a reproducible package would strengthen claims.
>
> **Answer :** We agree that consistent input protocols are crucial for fair comparison.
>
> **(1) Unified evaluation protocol.**
>
> Different baselines require different input modalities (e.g., prompts, masks, background handling), but we have made our best effort to unify all controllable inputs to ensure fairness. Specifically, we standardized resolution, camera viewpoints, number of views, and other shared parameters. The complete unified protocol is documented in Appendix A.2.
>
> Quantitative Results:
> | Methods    | FID ↓ | KID ↓ | Runtime ↓ |
> |------------|-------|-------|-----------|
> | Paint3D    | 71.32 | 7.35  | ~220s     |
> | MVPaint    | 50.94 | 4.32  | ~97s       |
> | TEXGen     | 44.82 | 2.96  | ~10s       |
> | MVAdapter  | 42.76 | 1.63  | ~33s       |
> | Ours(w/o character)       | 38.54 | 0.98  | ~4s       |
>
> **(2) Additional user study for fairness.**
>
> To provide an unbiased human-centered evaluation, we conducted a user study comparing our method and the baselines under the unified protocol. Following MVPaint’s evaluation criteria, the meshes are textured using our approach and all competing methods.
> Participants (N=5) were asked to evaluate:  1) Seamlessness 2) 3D Consistency using a 1–5 scale.
>
> The average scores are shown below:
>
> | Methods    | Seamless ↑ | 3D Consistency ↑ |
> |------------|-------------|------------------|
> | Paint3D    | 3.21        | 4.01             |
> | MVAdapter  | 3.87        | 4.62             |
> | MVPaint    | 3.71        | 4.05             |
> | TexGen     | 3.30        | 3.52             |
> | Ours       | 4.67        | 4.81             |
>
>
> **Weakness 2:** 3D consistency metrics are thin. The evaluation focuses on image-space metrics (e.g., FID/KID）and runtime. It lacks direct measures of UV seam continuity, cross-view consistency, and geometric adjacency color differences, which are crucial for textures that must look coherent on a mesh.
>
> **Answer :** To address this concern, we provide extensive multi-view qualitative comparisons showing that our method produces more coherent textures across seams and viewpoints than competing approaches (see Figure 5 and Figure 6 in the manuscript). Additional multi-view renderings (over 7 viewpoints) are included in Appendix A.7 to further support this observation. Moreover, following previous work, we conduct a user study that explicitly includes a criterion related to 3D consistency. Participants consistently preferred our method over the baselines on this criterion.
>
> #### User study results are shown above. We will make these points clearer in the revised manuscript.
>
> **Weakness 3:** Generalization beyond the training domain.  The dataset leans toward a specific visual domain. Claims of robustness to real photos, complex materials, heavy patterns, and challenging poses would be more convincing with systematic out-of-domain tests and a failure-mode analysis.
>
> **Answer :** To evaluate generalization beyond the training domain, we include several out-of-domain experiments in the supplementary material. Specifically, Appendix A.3 presents results on real-world photos, complex materials, heavy patterns, and challenging human poses. Appendix A.4 further provides qualitative results on the GarmentCodeData dataset, which has no overlap with our training data. In Appendix A.5, we also show cases where the garment mesh is not perfectly aligned with the reference image; even under such misalignment, our method can still produce reasonable textures.
> Across all these settings, our approach exhibits strong robustness and consistently good generalization performance. We acknowledge that our method may struggle with complex or irregular UV layouts, and we will expand the discussion of this limitation in the revised version.

---

### Author Response · Authors · 2025-12-03

Dear ACs,

We sincerely appreciate your time and effort in handling our submission, especially given the unusually heavy workload and the unexpected OpenReview issues this year. To facilitate a fair and efficient assessment, we provide a concise summary of the strengths identified by reviewers and the clarifications we provided during the rebuttal process.

**Positive Assessments from Reviewers**

Across the initial reviews, *GarmentPainter* received consistent recognition for:

* **Simple and practical design** with minimal modifications to the diffusion backbone (*ZtVi, 4r63*).
* **Sound and complete experimental validation** with convincing results (*GpsL, 4r63, 4FyR*).
* **Significance for industrial applications and the 3D community** (*GpsL, 4FyR*).
* **Valuable dataset contribution**, acknowledged by all reviewers.

 **Reviewer Concerns and Our Rebuttal Summary**

 **1. Fair comparison and additional metrics** (*ZtVi*)

We clarified evaluation settings and added two supplementary metrics—**Seamless** and **3D Consistency**—evaluated via user study.

 **2. Generalization ability** (*ZtVi, 4r63*)

We provided extensive additional results:

* **Appendix A.3:** real photos, complex materials, heavy patterns, challenging poses
* **Appendix A.4:** GarmentCodeData with no training overlap
* **Appendix A.5:** mesh–image misalignment cases

These demonstrate strong generalization across domains and UV layouts.

**3. Comparison with additional SOTA methods** (*GpsL, 4r63*)

We added comparisons against **MVAdapter** and **MVPaint**, where our method performs favorably.

**4. Implementation details and data split** (*GpsL, 4r63*)

We expanded training details, implementation specifics, and dataset splitting information to improve reproducibility.

**5. User study** (*GpsL, ZtVi*)

We conducted a user study evaluating **Seamless** and **3D Consistency** across five methods. Our method achieves the highest subjective ratings.

**6. Compatibility with stronger base models** (*GpsL, 4r63*)

We added experiments showing that GarmentPainter is **base-model agnostic**, and stronger backbones further improve performance.

**7. UV-based vs. multi-view-based texture generation**

We added a detailed discussion explaining why **UV-based generation** is more suitable for the garment texture task.

**8. Novelty clarification** (*4r63, 4FyR*)

We further clarified our contributions, including: **Unified Conditioning Strategy** and **Content-Aware Generation Without Strict Geometry Alignment**

**9. Type-selection module** (*4FyR*)

We clarified the motivation for this design and added additional ablations verifying its effectiveness.

**10. Automatic UV unwrapping** (*4r63, 4FyR*)

We explain that Automatic unwrapping destroys garment semantics by fragmenting islands and removing topology. Our method specifically targets structured, apparel-oriented UV layouts, which remain the dominant representation for real clothing assets and modern AI garment models. We will explicitly clarify this assumption in the revised version.

We thank the AC again for the time and effort dedicated to our submission. We believe we have thoroughly addressed all reviewer concerns, and we hope our responses help eliminate any remaining issues. We are confident that our method contributes meaningful insights to garment texture generation and offers practical value for industrial garment design.

Warm regards,

Authors of Submission 1933

---

### Meta-Review · Area_Chair_Vq3H · 2026-01-07

**Summary:**

This paper proposes GarmentPainter, a UV-space diffusion framework for 3D garment texture synthesis guided by a character image.

Reviewers acknowledge the method’s simplicity, efficiency, and potential industrial relevance, as well as the value of the curated dataset.

However, the overall consensus is negative due to concerns about limited novelty relative to prior UV-based texturing methods, insufficiently convincing evidence of generalization.

While the rebuttal adds experiments and clarifications, it does not fundamentally change these concerns. As a result, the paper does not meet the bar for acceptance.

**Reviewer Concerns:**

Concerns addressed by the rebuttal
- Added user study targeting seam smoothness and 3D consistency.
- Included comparisons with more recent baselines (e.g., MVPaint, MVAdapter).
- Clarified implementation details, data splits, and training protocol.
- Provided additional qualitative results on out-of-domain references and misalignment cases.

Remaining concerns
- Novelty: Using UV position maps and UV-space generation is viewed as incremental and closely related to prior work (e.g., Paint3D, TEXGen), with limited conceptual advancement.
- Generalization: Despite added experiments, reviewers remain unconvinced that results generalize beyond structured garment UVs and the curated dataset scale.
- Evaluation depth: Quantitative evaluation still relies heavily on image-space metrics and small-scale user studies, which are considered insufficient for strong claims about 3D consistency.

**Reviewer Scores:**

- Reviewer ZtVi: Remains 6 (borderline); acknowledges added experiments but remains unconvinced on evaluation depth and generalization.
- Reviewer GpsL: Keeps 6, the rebuttal addressed questions.
- Reviewer 4r63: Remains 2 (reject) due to strong concerns on novelty, dataset scale, and backbone choice.
- Reviewer 4FyR: Remains 4, move to reject; despite recognizing practical aspects, ultimately aligns with concerns on limited novelty and insufficient evidence to support the paper’s claims.

---

### Decision · Program_Chairs · 2026-01-26

Reject